# The Ibadan Hydrogeophysics Research Site (IHRS)—An Observatory for Studying Hydrological Heterogeneities in A Crystalline Basement Aquifer in Southwestern Nigeria

Kennedy O. Doro [1,*], Christianah O. Adegboyega [2], Ahzegbobor P. Aizebeokhai [3] and Michael A. Oladunjoye [2]

1   Department of Environmental Sciences, The University of Toledo, Toledo, OH 43606, USA
2   Department of Geology, University of Ibadan, Ibadan 200132, Nigeria
3   Department of Physics, Covenant University, Ota 231119, Nigeria
*   Correspondence: kennedy.doro@utoledo.edu; Tel.: +1-419-530-2811

**Abstract:** Crystalline basement aquifers are important drinking water sources in Nigeria and several sub-Saharan African countries. However, an understanding of their local flow and transport processes and pathways is missing due to limited research. The implication has been their suboptimal management, with frequently reported dry wells and groundwater contaminations. To address this challenge, the Ibadan Hydrogeophysics Research Site was established in 2019 as the first field-scale hydrogeological research laboratory in Nigeria to advance understanding of the geologic, hydraulic, and hydrogeochemical variabilities within crystalline basement aquifers. The over 22,500 m$^2$ research site with a 50 m × 50 m area used for active hydraulic testing is located within the University of Ibadan campus and is instrumented with four initial test wells extending through the weathered and fractured zones to a depth of 30 m each. Preliminary hydrogeological and geophysical studies focused on obtaining a conceptual model and knowledge of hydraulic heterogeneities to aid in detailed experimental and numerical studies. A combination of lithological logs and electrical resistivity revealed areas with subvertical fractures as low-resistivity zones (<200 Ωm), and a pumping test revealed a hydraulic conductivity range of $1.9 \times 10^{-10}$ to $7.2 \times 10^{-6}$ m/s. The drawdown–time curve shows flow from single-plane vertical fractures. The results of this study will serve as a basis for further targeted field and numerical studies for the investigation of variability in groundwater flow in complex crystalline basement aquifers. The presented field site is posed to support the adaptation and development of field methods for studying local heterogeneities within these aquifers in Nigeria.

**Keywords:** crystalline basement aquifers; groundwater; geophysics; southwestern Nigeria





## 1. Introduction

Crystalline basement aquifers constitute significant groundwater sources in tropical and subtropical regions in South Asia, South America, Australia, and sub-Saharan Africa [1–4]. These aquifers are formed by intense weathering and fracturing of impermeable igneous and metamorphic rocks, creating regolith of varying thickness, which overlies crystalline rocks with discontinuities such as fractures, joints, and faults [3–5]. Generally, their geometry, in a top-to-bottom sequence (Figure 1), can be subdivided into a topsoil layer, a weathered overburden termed saprolite, a weathered and fractured crystalline rock termed saprock, and a fresh crystalline rock serving as the aquitard [4,6–8]. Although basement aquifers are characterized by low groundwater storage and poor primary porosity and permeability, they are often locally important aquifers in locations where they are laterally extensive with significant overburden thickness and discontinuities [6]. In Nigeria, crystalline basement aquifers constitute major groundwater sources in the north–central, southwestern, and southeastern regions, with over 60 million people [9,10].

Groundwater flow and solute transport within crystalline basement aquifers are highly non-uniform, with complex local flow structures and high spatial and temporal variabilities in their hydraulic properties [11–14]. Their occurrence, geological and climatic controls, and hydraulic parameter distributions are yet to be fully understood, limiting their exploration and protection, mostly within developing countries such as Nigeria [9,15]. With a current population of over 200 million people in Nigeria, less than 70% of which have access to clean drinking water, there is increasing pressure to explore basement aquifers in the country [16–18]. However, to sustainably explore and manage these aquifers as drinking water sources, significant constraints limiting their use must be addressed [14]. These include (1) the difficulty in locating productive basement aquifers despite the recent increase in the integration of geophysical data; (2) frequent high borehole failure rates, with the higher rates in drier regions or areas with thin weathered overburden; (3) shallow occurrence and secondary permeability of the bedrock aquifer component serving as a conduit for infiltration and interaction with surface water, which makes them susceptible to surface pollutants; (4) the low storativity of basement aquifers, which depletes significantly during sustained drought periods; and (5) the high variability in their flow and transport properties, which is complicated by complex secondary porosity and permeability fields.

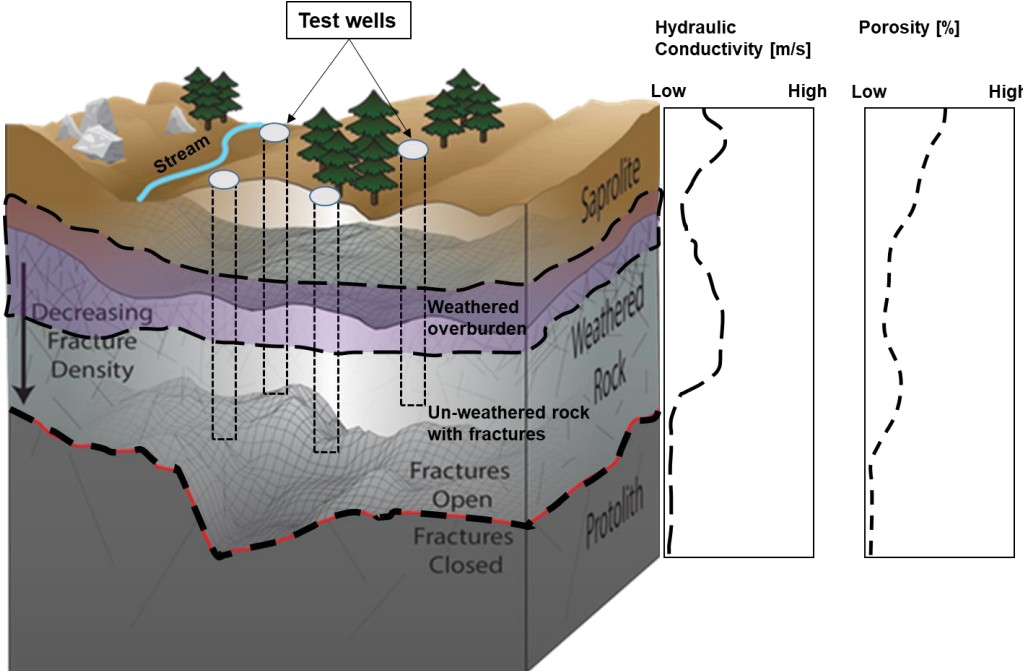

**Figure 1.** A conceptual model of crystalline basement aquifers and a qualitative hydraulic conductivity and porosity profile through their different layers (Adapted with permission from Flinchum et al. [19]. 2023, John Wiley and Sons.).

The Ibadan Hydrogeophysics Research Site (IHRS) was established in July 2019 to advance the current understanding of geologic heterogeneities and the hydrological processes within crystalline basement aquifers in southwestern Nigeria [20]. The field research facility will serve as a 3D controlled volume within a naturally heterogeneous crystalline basement aquifer to support research on adapting and developing geophysical and hydrogeological techniques to study the geologic, hydrologic, and geochemical variabilities within these aquifers. Earlier geophysical studies of basement aquifers in southwestern Nigeria and other parts of the country have been limited to using electrical resistivity, mainly one-dimensional (1D) electrical sounding and two-dimensional (2D) electrical resistivity tomography (e.g., [21–24]), as well as electromagnetics (e.g., [25,26]) and refraction seismic (e.g., [27]) methods, to delineate the aquifer zone. High uncertainties have been reported, with drilling results relying on predictions from these geophysical datasets, as has been

reported in results from other sub-Saharan African countries [28]. Hydrogeological testing and flow and transport modeling have also been reported at a coarse resolution incapable of resolving flow and transport properties and processes [29,30].

In this study, we present [1] an overview of the IHRS and [2] a combination of lithological, geophysical, and hydraulic test datasets to assess subsurface hydrological variability at the test site. A state-of-the-art review emphasizing research advances and knowledge gaps on crystalline basement aquifers in Nigeria highlights the need for a dedicated hydrogeological field research facility to study these aquifers in Nigeria. The results of initial geophysical and hydrogeological investigations are used to formulate a conceptual model of the site and to obtain a first understanding of its hydrogeological heterogeneities.

### 1.1. State of the Art

Groundwater flow within crystalline rocks has continued to attract global interest, especially the mode of occurrence [4–6,8,31,32], the origin of its porosity and permeability [3,33,34], and variabilities in its flow and transport properties [13,14,35–37]. It constitutes a significant source of drinking and irrigation water in tropical to subtropical regions, including Southeast Asia and sub-Saharan Africa [4]. Its high potential for connection to surface water through porous overburden, fractures, and faults serving as conductive pathways for flow and transport makes it highly susceptible to contamination [34].

The formation, structure, and occurrence of crystalline basement aquifers have been extensively discussed in the literature [4,6,38–42]. Their occurrence and properties are largely influenced by weathering, recharge, and secondary permeability in the crystalline rocks that allow for groundwater storage and flow [4,39]. Whereas crystalline basement aquifers are largely unconfined, they may also be semiconfined to confined aquifers in areas where the weathered and fractured basement is overlain by a very thick clayey regolith [43]. The collapse zone (topsoil), saprolite (weathered overburden), and the weathered portion of the saprock (Figure 1) are agreeably products of physical and biochemical weathering [3,4]. However, both tectonic fracturing and unloading [4], as well as weathering [3], have been hypothesized for the fractures and fissures in the bedrock, which drive flow in crystalline basement aquifers. Although a detailed discussion on the origin of fractures in crystalline basement aquifers is beyond the scope of this study, the occurrence of fractures in shallow crystalline aquifers has been generally observed to decrease with depth [43].

Crystalline basement aquifers are generally characterized by low storage and transmissivity, resulting in low aquifer yield [44]. They also strongly depend on recharge potentials, which vary with climatic conditions [4]. Although transmissivity is low, their hydraulic gradient is typically high, resulting in increased groundwater flux. Matrix hydraulic conductivity and effective porosity in crystalline basement aquifers are also low, whereas secondary porosity and fractured hydraulic conductivity can be significant but highly variable [43]. Such secondary porosity and hydraulic conductivity are controlled by fracture aperture, connectivity, and intensity [45]. Flow dominated by connected fractures can be considerably channelized following the dominant fracture planes within the aquifers, with the fractures constituting a small section of the rock volume [38]. Regions with high fracture connectivity with a corresponding high hydraulic conductivity result in advection-driven transport, whereas regions with poor connectivity and flow through their matrix are characterized by diffusion. Guihéneuf et al. [38] conceptualized the mass transfer process within crystalline basement aquifers to be dominated by advection, hydrodynamic dispersion, and matrix diffusion. However, they neglected the potential sorption and reactive terms. Understanding the dominant mass transfer process is critical for predicting the fate of solutes within these aquifers. However, this remains an active subject of research that is also currently being pursued at the IHRS.

As with other aquifer systems, predicting the recharge, groundwater flow, and solute transport within crystalline basement aquifers requires well-refined conceptual and numerical models [43,44]. Calibrating the numerical model requires field estimates of aquifer flow and transport parameters at a high spatial and temporal resolution. Drill cores

provide direct information on the lithology of the overburden and crystalline rocks and fractures within the basement rocks, which can be used to refine the conceptual model of the site. However, drilling and obtaining undisturbed cores in crystalline rocks are expensive, labor-intensive, and destructive, limiting their use. To obtain field estimates of flow and transport parameters, such as hydraulic conductivity, dispersivity, and porosity, well testing, including pumping, tracer, and flow meter tests, is used [8]. These tests have also provided insight into the fracture network, connectivity, and fracture- (advection) versus matrix (diffusion)-driven transport within these aquifers. Whereas the parameters estimated according to hydraulic tests are used to improve both conceptual and numerical aquifer models, such estimates are point-based or, at best, provide inter-well data limited in resolving variations within aquifers at a high spatial resolution and in three dimensions. Recent approaches have focused on using geophysical methods, including electrical, seismic, and ground-penetrating radar, to characterize the architecture and imaging fracture network and connectivity within crystalline basement aquifers [45,46]. This advancement improves the conceptual models of aquifers with 2D and 3D datasets [47]. Geophysical methods are also used to monitor aquifer tests, such as pumping and tracer tests [48,49], whereas the integration of geophysical and hydrogeological techniques for improving understanding of flow and transport within these aquifers is an active area of research [45].

Simulating flow and transport in crystalline basement aquifers is complicated due to the complex nature of the aquifer architecture resulting from a mix of connected fractures and rock matrix, which results in a non-uniform and anisotropic flow [50,51]. These heterogeneities and anisotropies are accounted for in a flow and transport model by adopting different approaches. These include representing the aquifer as a network of discrete fractures with matrix blocks of varying permeability; using a stochastic continuum with single, dual, or multiple components; or a combination of both approaches in a hybrid model with the stochastic continuum and statistical representation of the discrete fracture network [43]. Recent efforts to quantify flow and transport in these aquifers have focused on integrating geological, geophysical, and hydrogeological datasets, including isotopic data, to account for their highly heterogeneous and sometimes erratic behavior [43].

### 1.2. Research and Knowledge Gap in Nigeria

The use of groundwater in crystalline basement aquifers as a source of drinking and irrigation water in north–central, southwestern, and southeastern Nigeria has led to an increase in studies on such aquifers, with several documented peer-reviewed publications [15,20,21,23,29,52–61]. Despite this increased attention, there has been limited research advancing the understanding of the hydrological process within these complex aquifer systems. Most such research efforts are case studies [7] and mainly focus on characterizing the aquifer architecture to locate potential sites for drilling [26]. However, a few studies have explored remote sensing techniques for delineating potential aquifers [52–55], most of which are focused on using geophysical techniques to delineate potential aquifers, citing boreholes for domestic and industrial water supply [7,26,56]. A commonly used geophysical technique for groundwater exploration in Nigeria is the electrical resistivity technique with 1D electrical sounding (1D ES) to estimate overburden thickness and locate fractured zones [23]. Despite the inherent limitation of the 1D ES technique [28], its popularity can be attributed to the availability of low-cost equipment, ease of field implementation, and widely available open-source codes for processing and interpreting the data. Some studies have also explored the use of 2D electrical resistivity imaging (ERI) to characterize these aquifer systems [7,57]. Although 2D ERI can better capture the overburden thickness and fracture network distribution [7,28], it is less popular due to the lack of equipment and knowledge gaps in manually acquiring 2D data and using available open-source codes for data inversion. Whereas the number of case application studies using electrical resistivity continues to increase, only a few studies have focused on improving the adaptation of the method for local use considering the local geology. Adepelumi et al. [57] assessed the use of electrical resistivity imaging to delineate fractured zones in crystalline basement aquifers in

Nigeria and showed that the fractured zone is optimally detected when its width is at least half the overburden thickness. Some studies have also focused on adapting and developing new electrode arrays and assessing anisotropy within fractured aquifers [58]. Besides electrical resistivity, electromagnetic methods, including time-domain electromagnetic (TDEM) and very-low-frequency (VLF) electromagnetic [57], induced polarization [26], ground-penetrating radar, and refraction seismic [59] have also been used, although rarely, to delineate potential aquifers in the crystalline basement in Nigeria.

Only a few documented studies have focused on estimating hydraulic properties such as hydraulic conductivity, storage, and yield in crystalline basement aquifers in Nigeria [60,61]. Hydraulic conductivity ranging from $7 \times 10^{-4}$ to $2 \times 10^{-7}$ m/s has been reported using pumping tests [30,60,62,63]. However, these tests are primarily performed over a short duration and need to be more comprehensive to access potential boundary conditions. Several studies have also attempted to relate geophysical parameters (mostly electrical resistivity) to aquifer protection capacity and vulnerability [56,64] and hydraulic conductivity. These studies mainly relied on petrophysical relations from the literature. Geochemical characterization of crystalline basement aquifers in Nigeria has also been documented, mainly to assess groundwater quality and recharge sources [65–67]. Whereas most of these studies relied on analysis of chemical properties, including pH, electrical conductivity, and analysis of major cations and anions, some also used isotopic signatures to investigate residence time and provenance [67].

Although the value of a numerical framework for predicting groundwater flow and the fate and transport of solute within it has been emphasized, a limited number of studies have been reported for aquifer systems within the country, all of which are limited to sedimentary aquifers [29,68,69]. To the best of our knowledge, there has yet to be a documented study using numerical models to investigate groundwater flow within crystalline basement aquifers in Nigeria.

### 1.3. The Need for A Field Research Facility

The review of the state of research on groundwater occurrence and flow within crystalline basement aquifers in Nigeria presented in Section 1.2 shows that several studies have been conducted in an effort to understand aquifer architecture [6,57,64] and water quality [66,67]. Whereas these studies have advanced the exploration and use of groundwater in these complex aquifers, significant gaps exist in the knowledge of hydraulic parameter variations needed for the sustainable management of these aquifers. Such studies require the implementation of field experimental and numerical methods, which would benefit from a field-scale observatory for the development, adaptation, and implementation of such methods.

Field observatories for the study of groundwater systems, including the Boise Hydrogeophysical Research Site in Boise, Idaho, USA [70], and the Lauswiessen Test Site in Tuebingen, Germany [49,71], have been effective for the development of a predictive understanding of groundwater storage, flow, and transport and for the development of innovative field and numerical techniques [72]. Some field research facilities have also been dedicated to studying crystalline basement aquifers [73,74]. The Experimental Hydrogeological Park (EHP) in Choutuppal and the Maheshwaram basin test site established in southern India by the Indo-French Center for Groundwater Research (IFCGR) in 1999 have been very valuable in improving understanding of hydrological processes within these aquifers and for managing groundwater challenges in the region [74]. The AgrHys Observatories, including the Kervidy-Naizin and the Kerbernez catchments in western France, have studied shallow groundwater systems in an intensive agricultural region within a crystalline basement in a temperate climate [75]. A mesoscale observatory was also established in the subhumid Sudanian zone of the Upper Ouémé Valley in Benin, western Africa [74]. These observatories have been useful in understanding flow and transport within these aquifers, contributing to their effective management and use as irrigation and drinking water sources. Besides the IHRS presented in this study, there are no field

research observatories for the study of groundwater systems in Nigeria. Hence, the IHRS is poised to advance understanding of the occurrence and flow of groundwater within crystalline basement aquifers in southwestern Nigeria and the fate and transport of solute within them. It will also serve as a field laboratory for the development, adaptation, and validation of field methods for the investigation of crystalline basement aquifer systems. This will help to improve their management and ensure their sustainable use as drinking and irrigation water sources.

## 2. The Ibadan Hydrogeophysics Research Site (IHRS)

The Ibadan Hydrogeophysics Research Site is located within the crystalline basement complex of southwestern Nigeria. The site will serve as a field laboratory to advance understanding of hydrological processes within crystalline basement aquifers in parts of the country with similar architecture and serve as a guide for studies in regions where the crystalline basement aquifer has a significantly different architecture, e.g., a thicker weathered zone and a deeper active fractured zone. In this section, we first describe the regional geologic setting, followed by a local description of the study site.

### 2.1. Geologic Setting

Rock units within Nigeria are broadly divided into the basement complex, which underlies over half of the total area of Nigeria, and the sedimentary basins (Figure 2). The Nigerian basement complex forms part of the southern part of the Trans-Saharan, Pan-African mobile belt [76] of the Late Proterozoic (500–750 Ma) age. It lies between the Archean blocks of the West African Craton and the Congo Craton and the East-Saharan block to the northeast. The Nigerian basement complex extends westward, and it is continuous with the Dahomeyan of the Dahomey—Togo-Ghana region to the east and the south. Mesozoic recent sediments of Dahomey and Niger coastal basins cover the basement complex. The West African Craton and the Pan-African event, which represent the framework of West Africa in the entire igneous/metamorphic structural framework of Africa, consist of Precambrian rocks that have been subjected to major supracrustal plutonic events.

According to Rahaman [77,78] and Elueze [79], the basement complex of southwestern Nigeria where Ibadan is located is predominantly composed of migmatite and granitic gneiss; quartzite/quartz schist; slightly migmatized to unmigmatized metasedimentary schist and metaigneous rocks; charnockite; gabbro and diorite; and the members of the older granite suite, mainly granites, granodiorites, and syenites. Minor rock types found in the basement complex include pegmatite, which is composed of microcline and quartz; dolerite dykes associated with gneisses; and older granites, which are regarded as the youngest member of the basement complex (Figure 2).

Geologically, the University of Ibadan campus where the study site is located is underlain by quartzite, quartz schist, banded gneiss, and augen gneiss. Quartzite and quartz schist outcrops are found in the western part of the campus, whereas the eastern part is underlain by augen gneiss. Banded gneiss forms a strip between the quartzite and augen gneiss in a NW–SE direction. Quartzites, quartz schist, and augen gneiss outcrop in various parts of the campus, with the former being highly weathered, as evidenced by surface expression and foliation planes, which are well developed and strike in the NW–SE direction. These are part of Nigeria's Precambrian basement complex terrain comprising quartzites, banded gneisses, augen gneisses, and migmatites, whereas minor rock types include pegmatite, aplite, quartz veins, and dolerite dykes, as described by various authors [77,80,81].

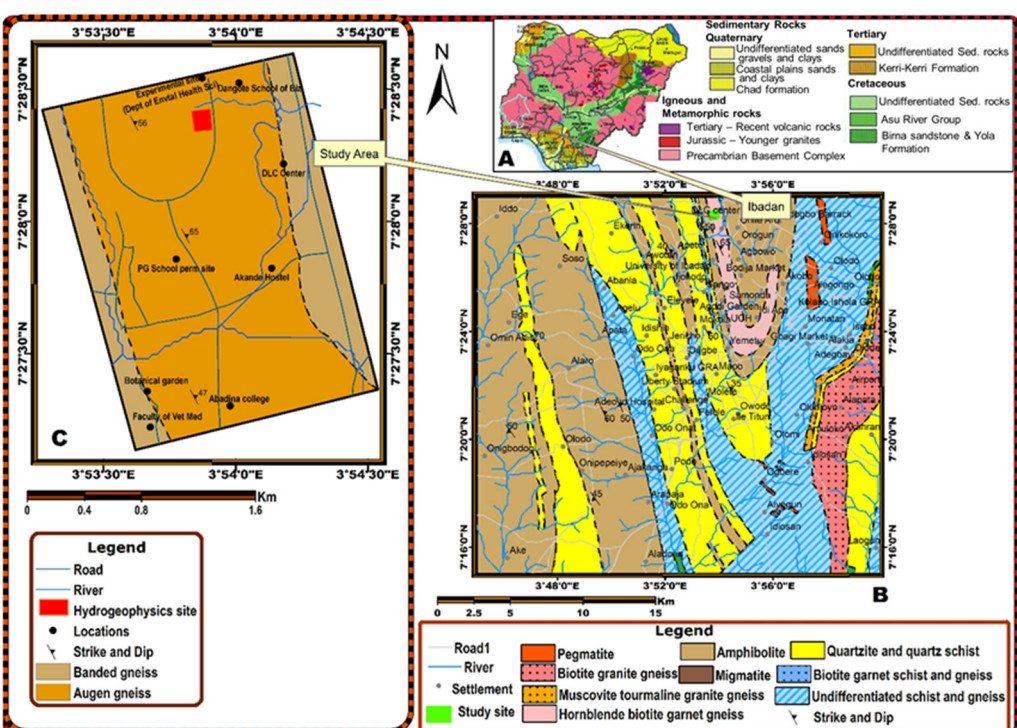

**Figure 2.** Geologic map of the study area showing the dominant rock units, their strike, and dip, as well as major rivers and roads (**B**). A map of Nigeria and an outline of the study area are shown in (**A**,**C**), respectively.

The study site is mainly underlain by augen gneiss and flanked in the eastern and western parts by banded gneiss trending approximately NW–SE. Field investigations showed that these rocks are outcropped in a few places with no observable surface expression of minor rocks. It is likely that the minor rocks are concealed within the older rocks, as evidenced by the drilling of experimental boreholes at the IHRS [20]. It is important to note that the augen gneiss observed on the surface outcrop at the site changes gradually to banded genesis at depth, as revealed by drilled cuttings. The relatively humid tropical climatic conditions aid the deep weathering of rocks, resulting in thick weathered profiles in most parts of southwestern Nigeria.

## 2.2. Site Description

The Ibadan Hydrogeophysical Research Site (IHRS) is situated in Ibadan, Oyo State, southwestern Nigeria (Figure 3). Specifically, it is located within the University of Ibadan campus extension along the Postgraduate School Permanent Site Road (Figure 3). The field research site comprises an approximately 31,000 m² area with an inner area of 2500 m² used for active hydraulic testing and monitoring. It is located within coordinates of latitudes 7°28′24″ to 7°28′26″ and longitudes 3°53′52″ to 3°53′54″ within a humid tropical climate with an average annual rainfall of 1240 mm [82]. The area around the site is characterized by a gently sloping topography with elevations ranging from 205 to 212 m above mean sea level. It stretches northward into a perennial stream flowing in the eastern direction and is flanked by the Dangote Business School Road eastward and by temporary farmlands (for agronomy students) south- and westward. The area is drained by a stream and linked to other smaller rivers. This has resulted in a trellis drainage pattern. The IHRS and surrounding area are undeveloped lands within the University of Ibadan's campus, where future research complexes are planned. The area is currently used as temporary farmlands with annual crops, including yam and maize, as well as biennial crops, such as cassava and pineapple. Besides the business school complex, which is about 500 m from the test site,

and a road parallel to the eastern boundary, the area is free of engineering structures and utilities.

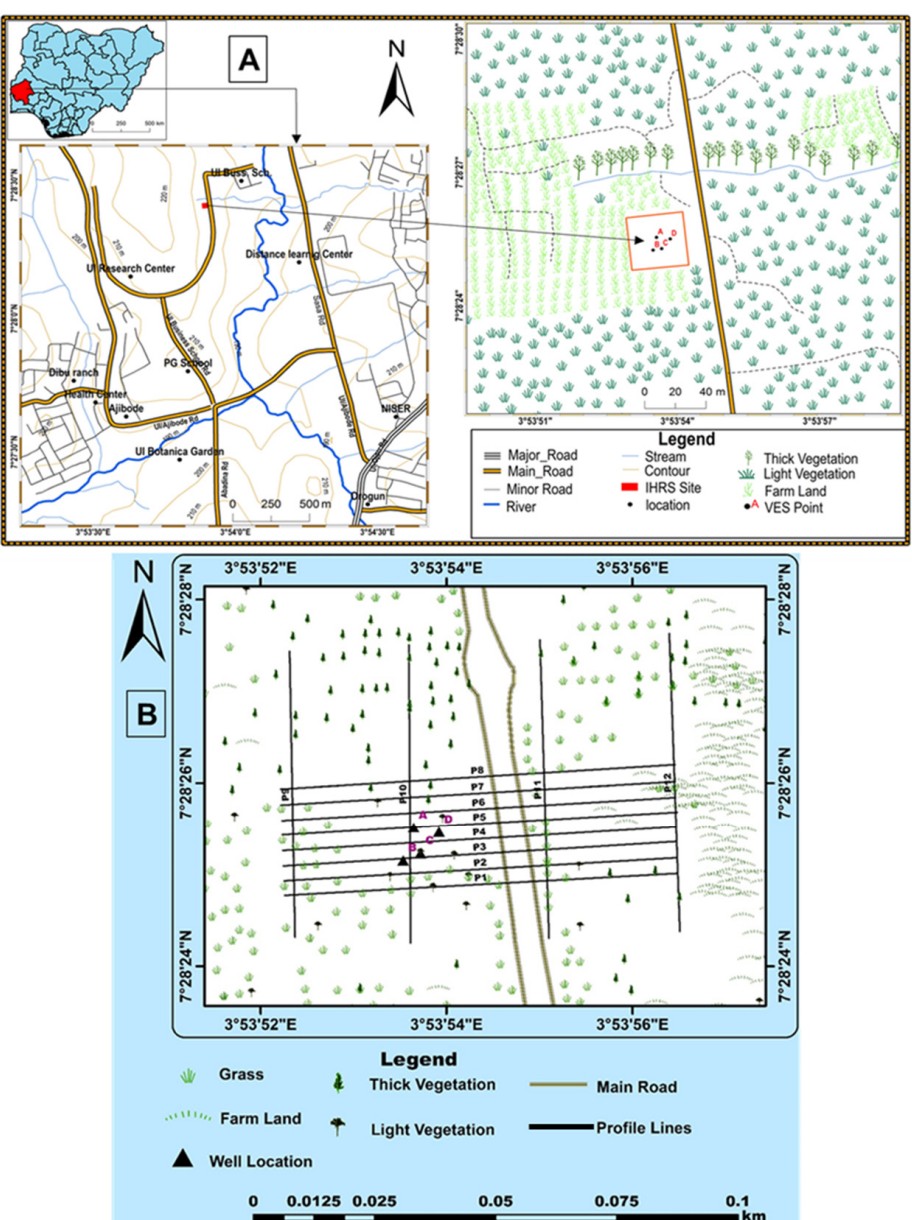

**Figure 3.** (**A**) Map showing the study site and the surrounding land use and (**B**) the locations of 1D electrical soundings and 2D ERT transects and boreholes.

## 3. Materials and Methods

### 3.1. Electrical Resistivity Survey

To obtain a first overview of the subsurface lithological variability at the research site, electrical resistivity measurements were conducted using a 1D electrical sounding (ES) approach. We injected current using a pair of electrodes and measured the resulting potential difference using a second pair of electrodes. Electrical resistivity soundings at increasing depths were achieved by increasing the current and potential electrode spacing following a standard approach described by Badmus and Olatinsu [22]. Six (6) electrical resistivity soundings were acquired using the Schlumberger array at locations corresponding to current well locations labeled A–D, as well as 10 m north and east of A (Figure 3). A Campus Ohmega—model 069 resistivity meter (Allied Associate Geophysical Ltd., Bedfordshire, UK) was used for data acquisition with standard minimum and maximum stacking of 3 and

6, respectively, and a maximum current injection of 1200 mA. Appropriate electrode contacts were established before each measurement with contact resistance below 2000 Ohms. Measured apparent resistivity data were plotted against the half-current electrode spacing (AB/2), and data processing was performed using Win-Resist, a 1D curve-matching resistivity computer code, to determine the geoelectric model parameters, including resistivity, layer thickness, and depth.

We also utilized 2D electrical resistivity tomography (ERT) to characterize the subsurface geologic variability at the site. Before field 2D ERT measurements, we first conducted a numerical study to assess the possible effect of the anticipated anomalies on measured resistivity, assuming two different scenarios, including subvertical fractures in the crystalline basement rocks and an inclined fracture plane according to approaches described by Alle et al. [28] and Adepelumi et al. [57]. A three-layer resistivity model accounting for the topsoil, saprolite (weathered overburden), saprock (fractured basement rock), and fresh basement rocks was generated based on earlier insight from 1D ES data and lithological logs. For simplicity, we modeled the saprolite and saprock as a single low-resistivity unit. Resistivity values were assigned to the synthetic model, and using Wenner, Schlumberger, and dipole–dipole electrode configurations, the distribution of apparent resistivities was calculated using a 2D finite-element-based forward solution, R2 resistivity code implemented in ResIPy [83,84]. These apparent resistivity values were later used as synthetic data after adding 5% Gaussian noise to simulate the field scenario and inverted using the Gauss–Newton-based inverse solution implemented in the same code [85,86] to generate the model response.

Guided by the preceding synthetic study, twelve (12) 2D electrical resistivity tomography (ERT) profiles were later acquired in this study (Figure 3) with a Campus Tigre resistivity meter–model 015 (Allied Associate Geophysical Ltd., Bedfordshire, UK) using a Wenner electrode array. Eight profiles with lengths of 130 m and spaced 4 m apart were acquired in the east–west direction, and four (4) profiles of the same length were acquired in the north–south direction. Unit electrodes were spaced at intervals of 2 m for all profiles. Measurement stacking, current injection, and contact resistance were similar to the earlier 1D measurements. Both forward and inverse resistivity modeling was carried out using the R2 resistivity code implemented in ResIPy [73], with the forward problem implemented by solving the Poisson resistivity equation in the Fourier transform domain [83]. In contrast, the solution to the inverse problem was based on the smoothness-constrained Occam inversion approach [86,87]. The eight (8) parallel 2D resistivity profiles running in the east–west direction were combined and inverted using a 3D resistivity inversion code to obtain a quasi-3D model of resistivity distribution at the site.

### 3.2. Well Installation and Lithological Logs

Based on preliminary results of 1D electrical sounding conducted at the test site, four test wells (A, B, C, and D in Figure 3) were installed for groundwater monitoring and hydraulic testing using an air rotary drilling technique. The test wells were located at the midpoint of the first four 1D–ES transects. Hence, the wells have the same location as 1D–ES 1–4 (Figure 3). The technique involves advancing a drill bit mounted at the end of a drilling pipe into the subsurface by rapid rotation, causing the drill bit to crush the rocks. Pressurized air is injected into the hole to cool the drill bit and to expel drill cuttings to the surface. Additional pipes were used to advance the drill bit to the desired depth of 30 m for each test well. Drill cuttings were obtained at 1 m intervals during drilling to create a lithological log of the wells. The choice of 30 m deep wells was based on drilling cost. Although deeper wells were desired, drilling was limited by prohibitive cost; hence, our choice to stop this first set of wells at 30 m provided a productive fracture zone for this preliminary study.

All wells were cased with 0.15 m diameter PVC pipes. The pipes were slotted to create well screens at a depth of 3–18 m extending through the weathered and fractured zone. Because caving in is not expected in boreholes in crystalline rocks, it is typical to case the

borehole only through the weathered overburden and the fractured rock zone. Developing wells by pumping regularly at high extraction rates is needed to prevent fine sediments from accumulating at the base of the well. After casing, the boreholes were gravel-packed using medium-grain gravels mostly consisting of quartz. The top 2 m were grouted to prevent direct infiltration through the sides of the boreholes.

*3.3. Hydraulic Testing*

A preliminary pumping test was conducted by extracting water at a rate of 1.7 L/s for 9 h at well A using a submersible pump. Changes in the hydraulic head with time (drawdown) were monitored simultaneously in the pumping well and at the monitoring well (C). Drawdown monitoring was performed manually using a water level meter at the extraction well (well A). Automatic monitoring was performed at well C using a CTD diver logger (Van Essen Instruments, Waterloo, Canada) installed in the well. We also monitored the atmospheric pressure using a barometric logger to correct for changes in atmospheric pressure during the pumping test. After shutting down the pump, recovery of the hydraulic head was also monitored in the pumping and observation well for 135 min.

We also performed a preliminary convergent forced gradient flow field and saline tracer test [51] by slug injection of a 400 L NaCl salt solution into well C while extracting water at 1.7 L/s at well D. Extraction was performed for 120 min at well D to create a near-steady-state scenario before tracer injection. Then, 5.2 kg of NaCl was mixed with 400 L of water to generate a tracer solution with a conductivity of 22 mS/cm compared to the ambient groundwater conductivity of 0.24 mS/cm. Changes in water conductivity were measured in situ in wells C and D using CTD diver loggers installed in each well.

## 4. Results

For this study, we combined direct lithological information obtained during well installation with the results of multidimensional electrical resistivity surveys to improve our understanding of the subsurface architecture of the study area. A combination of the subsurface architecture and results of the preliminary pumping and tracer tests were used to perform a first assessment of potential hydrogeological heterogeneities. We first present the lithological results and hydraulic tests in Section 4.1, which were used to constrain our interpretation of the electrical resistivity results presented in Section 4.2.

*4.1. Lithological Logs and Hydraulic Tests*

Lithological logs obtained from drill cuttings for all four (4) wells (wells A, B, C, and D in Figure 4) provide a direct estimate of the lithological distribution with depth at all four (4) drilling locations. Generally, all the logs show a top layer with a depth range of 1.5–3 m consisting of brown, silty, clayey sediments followed by a zone of weathered rock with sandy, clayey silt at a depth range of 5–8 m. These overburden materials show a coarsening downward sequence to a maximum depth of 8 m. The sediments are saturated at a depth of 5–6 m with a contact zone between the weathered overburden and the fractured zone observed between 8 and 14 m. During drilling, multiple fracture zones were encountered at 9, 14, 16, and 22 m, characterized by a high release of water, vein-filling material including fragments of quartz and feldspars, and distortion in the rate of penetration of the drill bit. A suspected fracture zone with quartz fillings was also encountered at a depth of 27 m in well B. Whereas fresh basement rocks were recovered at depths greater than 25 m in all the wells, it is unlikely that they penetrated all opened fractures. It is safe to assume that the drilling was stopped within the active fracture zone.

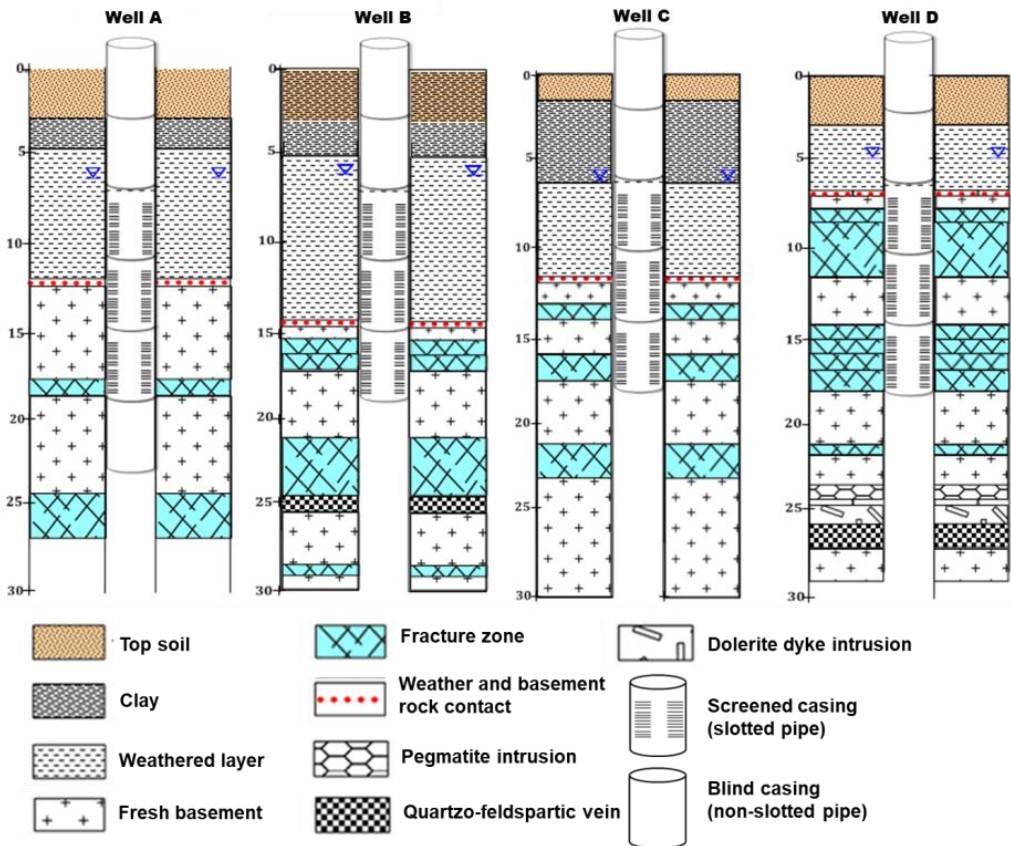

**Figure 4.** Lithological profiles and well design for all four wells (wells A, B, C, and D). The lithological logs were created from drill cuttings obtained every 1 m or when changes occurred in sediment types during drilling.

The well design for the four (4) wells installed at the site is also shown in Figure 4. All the wells were drilled to 30 m and cased with 0.15 m diameter polyvinyl chloride (PVC) pipes. The well casing extends to 18 m, which corresponds to the top of the fractured hard rock. The PVC pipes used to case the wells were slotted at depths of 3–18 m to create a screened segment. Screening the well within the hard rock region (18–30 m) is typically not required, as well collapse is minimal in this region and allows for direct contact with the fractures, which serve as water conduit paths into the well.

The static groundwater level measured in the wells is approximately 201.4 m above mean sea level, and the flow is in the northeast direction and towards a stream at the site. The pumping test created a drawdown of 3.9 m in the pumping well (well A) and 3.6 m in well C, which was used as an observation well. Drawdown–time plots on semi-log and log–log plots are shown in Figure 5A,B, respectively. None of the wells had complete recovery within the 135 min of post-pumping recovery monitoring. The pumping test data were fitted using the Baker analytical solution for fractured aquifers [88,89] (Figure 5C) and were used to obtain estimates of fracture and matrix hydraulic conductivities of $7.2 \times 10^{-6}$ and $1.9 \times 10^{-10}$ m/s, respectively. The injected salt tracer at well C with electrical conductivity of 22 mS/cm compared to the background groundwater electrical conductivity of 0.24 mS/cm did not produce any measurable change in the electrical conductivity at well D.

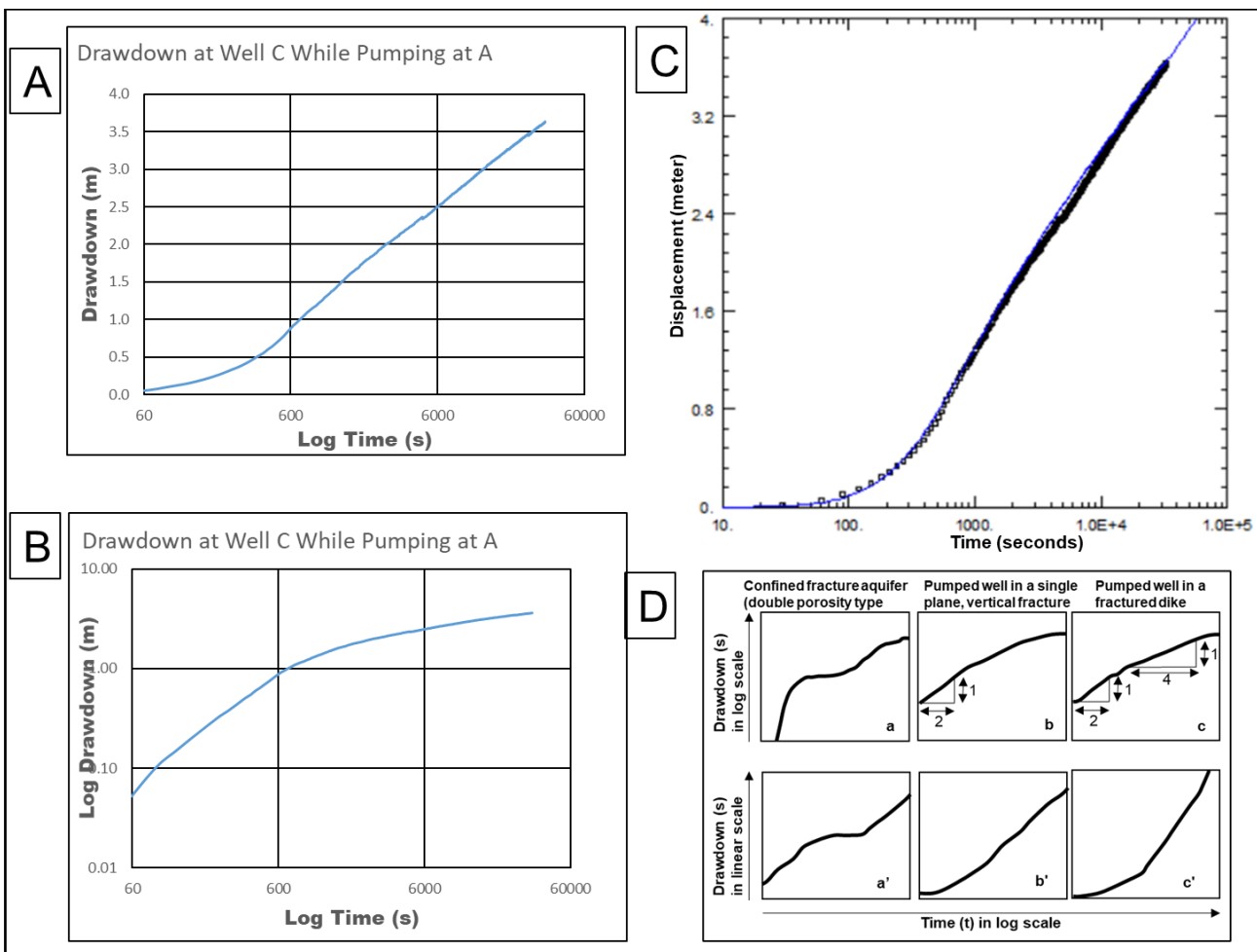

**Figure 5.** A drawdown versus time plot on a semi-log (**A**) and log–log (**B**) plot for observation at well C while pumping at well A. A fitted drawdown–time curve of the pumping test using the Baker solution for fractured aquifer [88,89] (**C**) with type curves [90] for interpretation of the results are (**D**).

*4.2. Electrical Resistivity*

The 1D models obtained from the electrical sounding (ES) data show resistivity variation with depth at a specific location corresponding to the center of the ES transects. Results of selected 1D resistivity models are presented in Figure 6, and a summary of the 1D ES data is shown in Table 1. Apart from ES 1, which shows a three-layer apparent resistivity distribution curve, all other ES apparent resistivity distribution curves show a four-layer model. The number of layers is inferred from visual inspection of the curves based on inflection points on the curves and the 1D inversion model output of the various layers, their resistivity, thickness, and depths for each sounding curve. Generally, the 1D models show a relatively resistive and thin top layer with resistivity ranging from 41 to 573 $\Omega$m. The three underlying layers generally show an increase in resistivity with depth, with the last layer having a resistivity greater than 1000 $\Omega$m. We compared the 1D ES results with the lithological logs obtained by drilling to interpret the various resistivity layers (see Table 1). The four resistivity layers can be correlated to the topsoil, weathered overburden, fractured basement, and fresh basement rocks. The high-resistivity range of the topsoil (41 to 573 $\Omega$m) is reflective of the varying clay content. Areas with higher clay content correlate with low resistivity. The increase in resistivity with depth could be related to a decrease in weathering and fracturing intensity.

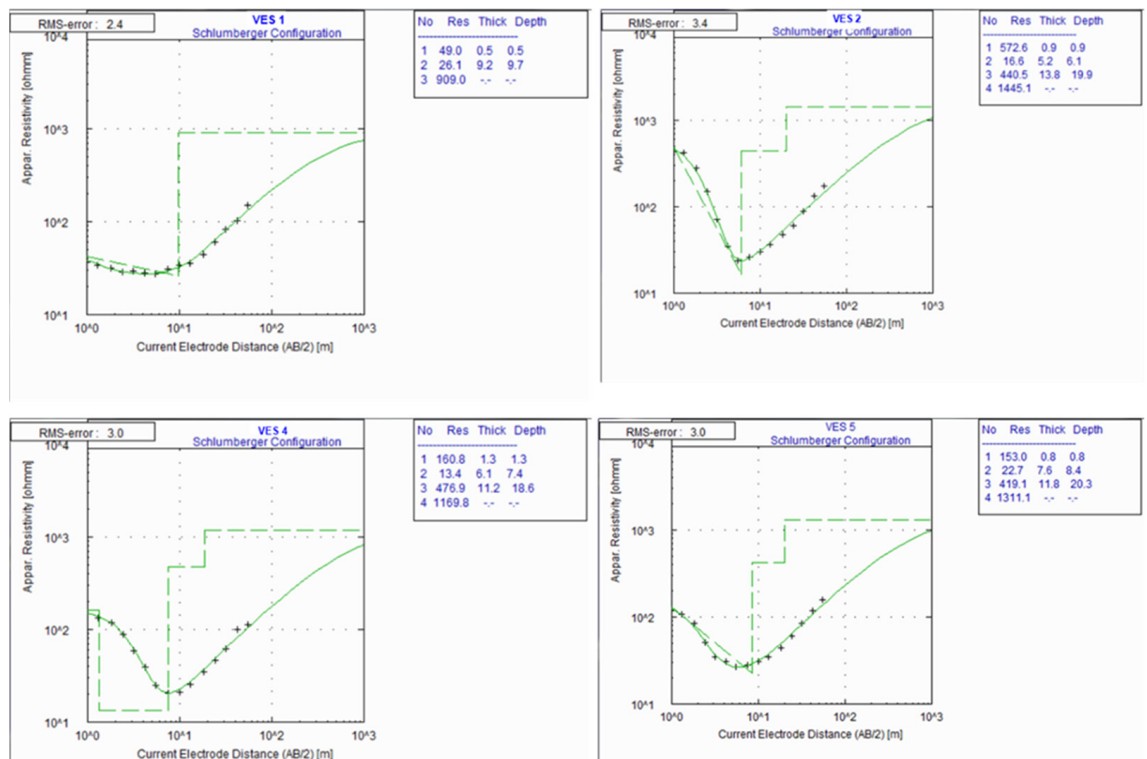

**Figure 6.** One-dimensional electrical resistivity sounding curves and models for VES 1 and 4 (**left**) and VES 2 and 5 (**right**). The model parameters, including the number of layers, resistivity, thickness, and depth, are shown to the right of each curve.

**Table 1.** A summary of the 1D electrical sounding results at all six sounding stations with the numbers of layers, resistivities, thicknesses, and depths extracted from the model results. The lithology was interpreted based on a comparison with the lithology from well logs.

| VES Station | Layers | ρ (Ωm) | h (m) | d (m) | Lithology |
|:---:|:---:|:---:|:---:|:---:|:---:|
| | 1 | 41 | 0.6 | 0.6 | Topsoil |
| 1 | 2 | 27 | 9.3 | 9.9 | Weathered layer |
| | 3 | 847 | - | - | Fractured basement |
| | 1 | 573 | 0.9 | 0.9 | Topsoil |
| | 2 | 17 | 5.2 | 6.1 | Weathered layer |
| 2 | 3 | 444 | 13.4 | 19.5 | Weathered/fractured basement |
| | 4 | 1302 | - | - | Fresh basement |
| | 1 | 480 | 0.5 | 0.5 | Topsoil |
| | 2 | 83 | 0.9 | 1.4 | Weathered layer |
| 3 | 3 | 13 | 4.8 | 6.3 | Weathered/fractured basement |
| | 4 | 1344 | - | - | Fresh basement |
| | 1 | 162 | 1.3 | 1.3 | Topsoil |
| | 2 | 14 | 6.3 | 7.6 | Weathered layer |
| 4 | 3 | 467 | 11.1 | 18.7 | Weathered/fractured basement |
| | 4 | 1073 | - | - | Fresh basement |

**Table 1.** *Cont.*

| VES Station | Layers | ρ (Ωm) | h (m) | d (m) | Lithology |
|:---:|:---:|:---:|:---:|:---:|:---:|
| 5 | 1 | 153 | 0.8 | 0.8 | Topsoil |
| | 2 | 23 | 7.5 | 8.3 | Weathered layer |
| | 3 | 402 | 13.1 | 22 | Weathered/fractured basement |
| | 4 | 1279 | - | - | Fresh basement |
| 6 | 1 | 79 | 1.2 | 1.2 | Topsoil |
| | 2 | 12 | 4.2 | 5.5 | Weathered layer |
| | 3 | 465 | 15.7 | 21.2 | Weathered/fractured basement |
| | 4 | 1554 | - | - | Fresh basement |

ρ is the estimated average layer resistivity, h is the thickness of each resistivity layer, and d is the depth from the top to the bottom of the layers. Lithology interpretation is based on the lithological profiles of the well drilled at the site.

The results of our forward model obtained before the field investigation are presented in Figure 7. All three electrode arrays tested in the numerical study (rows 2–3 in Figure 7) capture the three-layer models (row 1 in Figure 5), with the inversion enforcing a smoothening across the layers. The subvertical fracture zone represented by low-resistivity depressions into the high-resistivity bedrock in the conceptual models are captured as low-resistivity zones in the model results. Inversion results of synthetic data with the Wenner array show a higher sensitivity to the subvertical fracture zone, whereas the dipole–dipole array shows the least sensitivity. A simulated inclined fracture plane with a 4 m width represented in the conceptual model by a low resistivity (row 1 (right) in Figure 7) is not captured in the model by any of the electrode arrays.

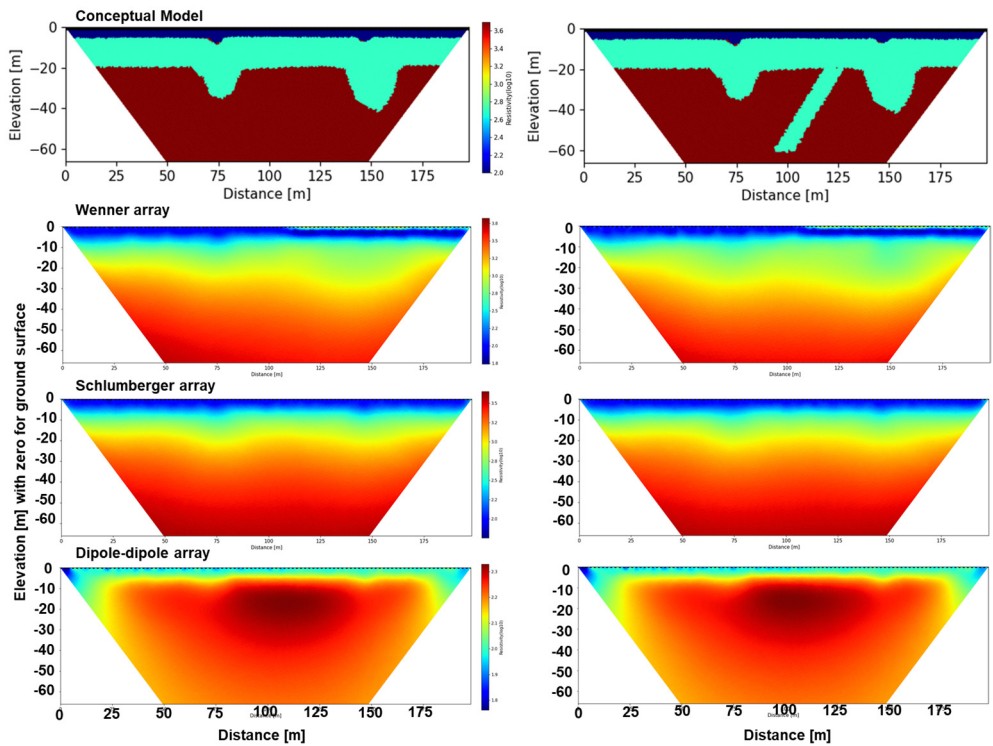

**Figure 7.** Results of forward simulation with a three-layer conceptual model with a 2 m wide subvertical fracture (**right**) and without fracture (**left**) in the top row. Inversion results from synthetic data obtained from a forward simulation using this model representation are shown in rows 2, 3, and 4 for Wenner, Schlumberger, and dipole–dipole electrode arrays, respectively.

The 2D electrical resistivity imaging (ERI) results show the variation in resistivity across a profile with depth, better capturing the subsurface heterogeneity. Selected images of 2D resistivity models for profiles running east–west are shown in Figure 8. Generally, the resistivity models converged after approximately four iterations, with root mean square error (RMS) values ranging from 1 to 2%. The modeled resistivity shows four distinctive resistivity zones labeled 1–4 in Figure 8. The 2D resistivity distribution is similar to that of the 1D ES. Besides the top resistivity layer with high resistivity (~1000 Ωm), the resistivities of the second to fourth layers (2–4) increase with depth. A fifth zone of low resistivity (~75 Ωm) labeled 5 is identified within the third resistivity zone with medium resistivity (250 Ωm) in rows 1 and 2 of Figure 8. To assess existing trends in the resistivity distribution, the results of all resistivity profiles in the east–west direction and those in the north–south direction were plotted in fence-like diagrams (Figures 9 and 10, respectively), revealing variation in the depth to the fourth resistivity layer with high resistivity. Additionally, the occurrence of low-resistivity zones is highlighted by red dashed ovals.

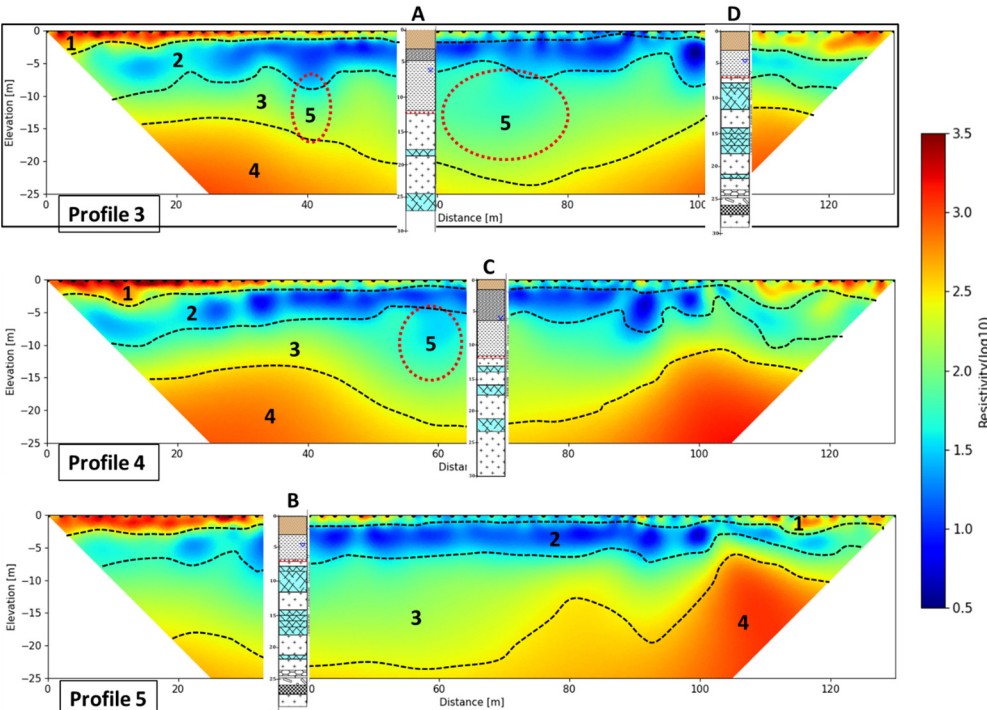

**Figure 8.** Selected subsurface resistivity models from inversion of ERT data for (**top**) profile 3, (**middle**) profile 4, and (**bottom**) profile 5. Lithological logs for wells A–D are placed on the corresponding resistivity profiles.

We placed the lithological logs at their corresponding locations along the resistivity profiles (Figure 8) to interpret the 2D resistivity distribution. Wells A and D are on profile 3, well C is on profile 4, and well B is on profile 5 (Figure 8). In all cases, the topsoil consisting of clayey sand correlates with high-resistivity zone 1. Areas with higher resistivity have higher sand contents, which is consistent with the 1D ES results and field observations. The low- and medium-resistivity zones (2 and 3) correlate with the clayey silt of the weathered layer and the fractured bedrock. High-resistivity zone 4 is interpreted as fresh bedrock or a region of the bedrock with few or no fractures. Regions with low resistivity within the fractured bedrock correlate with areas where fractures were observed, which are highlighted by red dashed ovals in the figure. These are potential areas with higher amounts or concentrations of subvertical fracture networks.

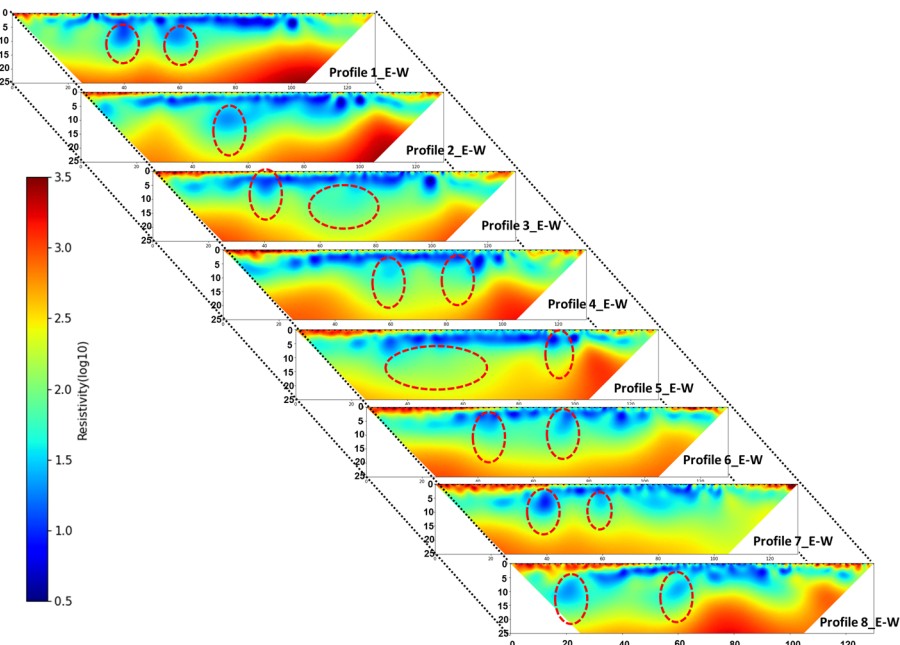

**Figure 9.** Resistivity models along transects 1 to 8 acquired in the east–west direction with profiles 6, 7, and 8 towards the stream at the northern flank of the site.

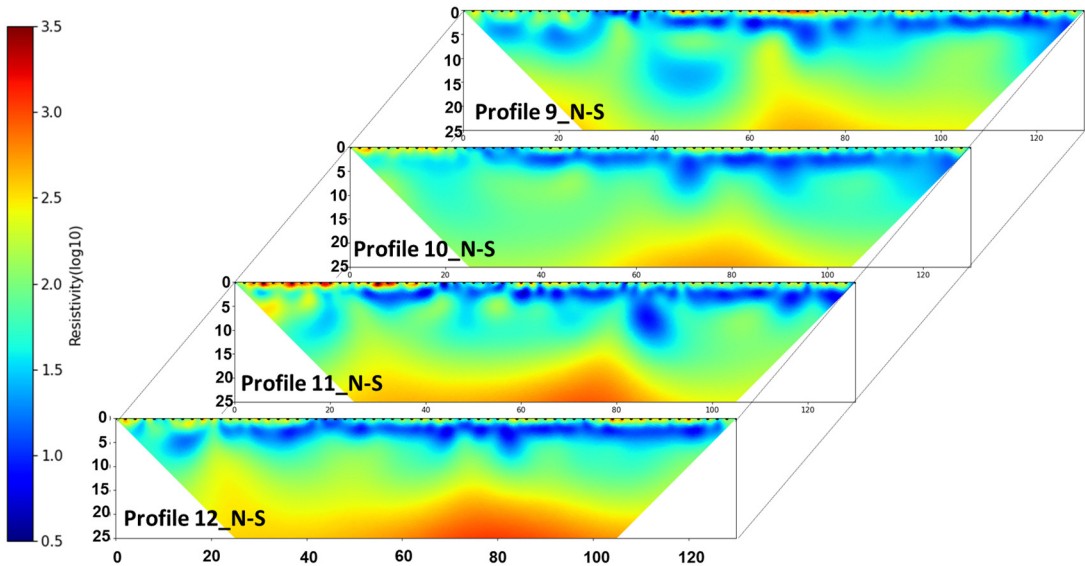

**Figure 10.** Resistivity models along transects 9 to 12 acquired in the north–south direction.

A quasi-3D resistivity distribution model was created by combining all eight parallel 2D profiles in the east–west direction and inverting them jointly using a 3D inversion routine implemented in ResIPy software [83]. Because the 2D transects were acquired using a 2 m unit electrode spacing with the transects spaced every 4 m, combining the transects is possible and according the rule of the profile spacing not being more than twice the unit electrode spacing [47,84]. The 3D model inversion, which converged after the sixth iteration with an RMS value of 6.9%, is presented in Figure 11 to highlight the spatial variation in resistivity at the site. The quasi-3D resistivity distribution is, not surprisingly, similar to the 2D results. The high-resistivity top layer appears to be more prominent in the eastern and western segments of the site (Figure 11). The 3D inversion result also reveals variation in the bedrock topography.

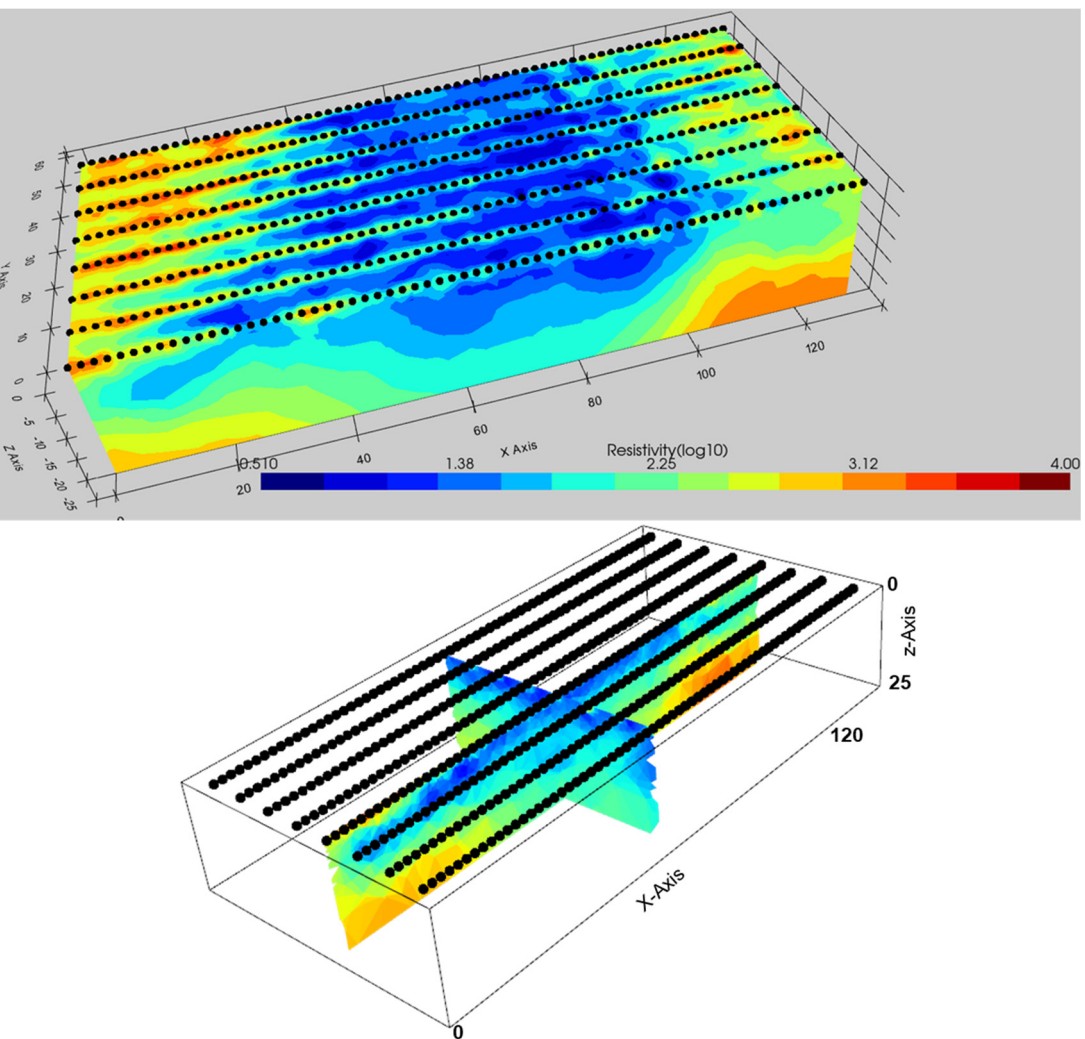

**Figure 11.** 3D resistivity model obtained by combining 2D measurements along profiles 1 to 8.

## 5. Discussion

Characterizing subsurface geologic and hydraulic heterogeneities within crystalline basement aquifers is challenging [5,7]. Drilling, obtaining core samples, and installing test wells through crystalline basement rocks are difficult and expensive and are limited in capturing spatial variation in hydrogeological properties [20]. Low-cost 1D geophysical techniques used in Nigeria, e.g., electrical sounding popularly, have also been shown to be limited [28]. In this study, we combined lithological logs prepared from drill cuttings obtained while drilling four preliminary test wells at the site with multidimensional electrical resistivity imaging to characterize the aquifer architecture and structural variability. We also used a preliminary pumping and tracer test to gain insight into the aquifer's hydraulic properties. Here, we discuss the results of this study and use them to show how the IHRS can be expanded to improve understanding of hydrological processes within crystalline basement aquifers in southwestern Nigeria.

### 5.1. Architecture and Heterogeneity of the Crystalline Basement Aquifer at IHRS

Combining the results of 1-, 2-, and 3D electrical resistivity distributions (Figures 6–11) with lithological logs of wells drilled at the site (Figure 4) allowed us to better characterize the aquifer architecture. The top layer is generally composed of silty and sandy clays, indicating the chemical weathering of the crystalline rocks in the area [4,91]. The top layer (regolith) with high clay content and significant thickness (>3 m) overlying the weathered and fractured crystalline basement aquifer at the IHRS may limit infiltration and reduce

aquifer recharge [92]. However, it may also provide an improved protective capacity for the underlying aquifer [65]. The weathered zone (saprolite) is presumed to exist within the second layer, as seen in the geoelectric model as lower resistivity and in the lithological logs. The weathered layer is mainly composed of clayey and silty sands, fine gravel, and pebbles, which are products of in situ weathering of the augen gneiss bedrock in the region. According to drill cuttings, the weathered layer contains mica minerals, which are indicative of potential weathering of banded gneiss. The weathered zone (saprolite) is extensive across the site and delineated as a low-resistivity zone (5 Ωm to 80 Ωm), with a depth range of 3 m to 9.7 m. The weathered zone is saturated and serves as part of the aquifer system. However, the groundwater yield of this layer is a function of the regolith thickness, the clay volume, and the interconnectivity of the grains [43,93]. The resistivity results (Figures 6–11) only extend to 25 m but show the resistivity distribution of the upper part of the fractured basement. Regions below 10 m in the 2- and 3-D resistivity models (Figures 6–11) with low resistivities (5–200 Ωm) are interpreted as areas with a highly fractured network that are potentially productive fractured aquifers. However, the resistivity models are limited in isolating the individual fractures that were observed while drilling.

The 1D ES results helped to estimate the thickness of the weathered and fractured zones, which is critical in predicting the potential locations for water wells. However, the 1D ES models underestimated the thickness and depth of the weathered overburden and fractured basement. Hence, this method should be used with caution when solely used to delineate water-bearing layers within the crystalline basement. This is consistent with the results of an earlier study by Alle et al. [28]. The 2D resistivity models from ERT (Figures 6–11) provided a better understanding of subsurface heterogeneity at the site, particularly areas with a higher concentrations of fractures interpreted as low-resistivity zones within high-resistivity basement rocks. However, the contrast between the weathered overburden and the fractured basement (described as zone 2 in Figure 8) is less evident in some areas. A comparison of the result of the synthetic study with that of the field study shows that interpretation and understanding of such contrast can be greatly improved when the 2D resistivity model interpretation is guided by knowledge of the anticipated anomaly effect on the resistivity distribution. This confirms the results of earlier numerical studies by Adepelumi et al. [57]. However, such results rely on an a priori information, which, in this study, was available through drilling and initial VES measurements. It is also worth emphasizing that although resistivity models can locate areas with a high subvertical fracture network as relatively low-resistivity zones, they cannot isolate the individual fractured zones encountered while drilling. Groundwater yield in the fractured zones depends on the density and connectivity of the fractures; hence, mapping highly fractured zones mostly as low-resistivity regions would help to improve the conceptual hydrological understanding. The fresh, unweathered basement considered in this study was only inferred from the 1D and some 2D resistivity models, with resistivity ranging from 910 Ωm to 3620 Ωm. The unweathered and unfractured basement constitutes the aquitard in the area.

*5.2. Hydraulic Properties of the Crystalline Basement Aquifer at IHRS*

Based on the resistivity models and lithological logs, the aquifer at the study site consists of clay-rich weathered zone sediments in the upper 3–10 m (saprolite), as well as fractured augen and Banded gneiss (saprock), which likely extend beyond the study depth of 30 m. The aquifer is conceptualized as an unconfined aquifer with a thickness of more than 25 m. The resistivity models were constrained to only 25 m depth due to a limited profile length, and the wells were constrained to a depth of only 30 m due to limited funding and the associated high cost of drilling. Therefore, the wells extend through approximately 15 m of the fracture zone. We anticipate that the active fracture zone extends further, given suspected fractures at a depth of 29 m observed in well B.

The average fracture and matrix hydraulic conductivities were estimated as $7.2 \times 10^{-6}$ and $1.9 \times 10^{-10}$ m/s, respectively, based on a pumping test representing integrated values for both the weathered and fractured zones. A low hydraulic conductivity typically characterizes the weathered compared to the fracture zone with secondary connectivity. Based on the measured drawdown during pumping at well A, well C appeared to be hydraulically connected to A, with an appreciable drawdown in well C (3.6 m) induced by pumping in well A. The comparison of the semi-log and log–log plots of the drawdown with time (Figure 5A,B) and type curves by Kruseman et al. [90] presented in Figure 5D was used to assess the boundary conditions and water supply region. The drawdown observed in this study is similar to the type curve for pumping from wells in single-plane vertical fractures (middle plots in Figure 5D). The log–log curve shows an early time slope close to half that approaches a plateau at a later pumping time. The semi-log plot shows a rising slope at a late time. Although this result is inconclusive due to the limited pumping duration of 9 h, this interpretation is consistent with the existence of subvertical fractures conceptualized for most crystalline basement aquifers [3,5,7,19,39]. Although the existence of incline fracture planes cannot be ruled out, further hydraulic tests are required to confirm their presence.

The preliminary tracer test conducted during this study was unsuccessful. The 400 L of saline water with an electrical conductivity of 22 mS/cm injected at well B with continuous extraction at well D to test potential well connectivity did not produce any measurable changes in electrical conductivity at well D. This raises questions about the potential connections between the fractures, especially between wells B and D. Knowledge of fracture continuity and connectivity within crystalline basement aquifers is needed to understand flow and transport. However, delineating them remains a continuing challenge in fractured aquifer studies.

*5.3. Expanding the IHRS and Further Research*

The IHRS is the first field research facility to study groundwater flow and transport in crystalline basement aquifers in Nigeria. The site is expected to stimulate further research to advance the understanding of hydrological processes within these complex aquifer systems. Although the site has been instrumented with four test wells and preliminary geophysical and geological data have been acquired, the wells have a limited depth of 30 m. The electrical resistivity models are also limited to a depth of investigation of fewer than 25 m due to limited equipment to acquire geophysical data across longer transects. Despite these limitations, these datasets form a basis for extending the infrastructure and research at the site.

Future electrical resistivity studies will focus on transects > 500 m to obtain models with a depth of investigation > 100 m. Further geophysical surveys using refraction seismic and electromagnetic imaging are also planned to better characterize the weathered layer and bedrock topography. Deeper test and monitoring wells are planned to target possible deeper fractures at depths > 100 m. Additional test and monitoring wells are also planned to be installed in a nested format targeting the weathered and fractured layers separately. This will allow for characterization of the hydraulic properties of these layers separately. The research site and infrastructure are open to both local and international collaborators interested in advancing understanding of the hydrological processes in crystalline basement aquifers.

## 6. Conclusions

Understanding flow and transport variability within crystalline basement aquifers remains a challenge globally and in Nigeria. Hence, a field research laboratory (the IHRS) was established to advance knowledge of hydrogeological heterogeneity within these aquifer systems. A review of the state of the art and research conducted to date in Nigeria was used to highlight gaps in understanding of hydrological processes in crystalline basement aquifers. Significant advances have been made in the use of geophysical and

hydrogeochemical methods to delineate aquifer architecture and evaluate water quality in Nigeria [7,10,21–27,56,58,66,67]. However, few field experiments have been conducted involving pumping and tracer tests, numerical modeling, adaptation and development of field methods, and integration of geophysical and hydrogeological methods reflecting the state of the art to characterize state variables and processes within these aquifers. The establishment of the IHRS as a dedicated field facility to study crystalline basement aquifers in southwestern Nigeria is expected to stimulate research advances. The site is open for local and international research collaboration to advance the understanding of groundwater storage, flow, and transport within these aquifers.

The initial hydrogeological and geophysical datasets obtained in this study confirm the earlier conceptual model of crystalline basement aquifers made up of clay-rich topsoil and saturated weathered overburden overlying a fractured basement rock [3–6,19,39]. The weathered overburden at the test site shows a varying thickness (7.5–14 m), with thinner portions in the eastern and western segments of the site (Figure 8). The weathered overburden is also saturated at a depth of ~5 m. The delineated fractured bedrock shows an undulating topography with several primary fracture successions at 9, 14, 16, 22, and 27 m. The aquifer is conceptualized as an unsaturated aquifer system with clay-rich topsoil and overburden sediments that could limit infiltration but enhance protection from surface containments. A combination of lithological logs (Figure 4), resistivity models (Figure 8), and hydraulic tests (Figure 5) shows that the aquifer is highly heterogeneous, with variation in the overburden thickness, bedrock topography, and the distribution of subvertical fractures and hydraulic conductivity and connectivity. Understanding the spatial distribution of these heterogeneities and assessing their impacts on flow and transport is currently an ongoing research focus at the site. This and subsequent studies will improve the current understanding of the 'state', hydraulic parameter distributions, and hydrological processes needed to sustainably utilize and protect crystalline basement aquifers as drinking water sources in Nigeria. A detailed pumping and tracer test is currently being implemented at the site to estimate the distribution of hydraulic parameters. A detailed groundwater flow and transport model is also being developed to improve the prediction of groundwater recharge, storage, flow, and transport.

Although electrical resistivity techniques are useful in delineating the thickness of the weathered overburden and areas with intense fracturing, they are limited in imaging individual fracture planes. This is not unconnected with the "smoothening effect" associated with resistivity inversion [94]. We propose further geophysical investigation using electromagnetic imaging and cross-hole ground-penetrating radar to image the fracture planes.

This study also highlights the current approach of characterizing fractured crystalline aquifers in Nigeria as a single hydraulic unit. This limits understanding of the contributions of the weathered zone assumed to have low permeability compared to the fractured zone with high permeability. A future study is planned at the site to isolate the hydraulic properties of the weathered layer from the fractured zone by conducting hydraulic testing, including pumping and tracer tests using multilevel packer systems to isolate stress–response effects in each zone. A separate set of tests and monitoring wells is also planned to target only the weathered overburden. The current setup of the site with the test wells extending only through the active fracture zone (30 m depth) due to funding constraints is limiting. Future studies will involve drilling deeper wells based on geophysical investigations with a depth of investigation > 100 m.

**Author Contributions:** Conceptualization, K.O.D. and M.A.O.; methodology, K.O.D., A.P.A. and M.A.O.; software, K.O.D.; validation, K.O.D., C.O.A., A.P.A. and M.A.O.; formal analysis, K.O.D., C.O.A. and M.A.O.; investigation, K.O.D., C.O.A., A.P.A. and M.A.O.; resources, K.O.D. and M.A.O.; data curation, K.O.D., A.P.A. and M.A.O.; writing—original draft preparation, K.O.D.; writing—review and editing, K.O.D., C.O.A., A.P.A. and M.A.O.; visualization, K.O.D., C.O.A., and M.A.O.; supervision, K.O.D. and M.A.O.; project administration, K.O.D. and M.A.O.; funding acquisition, K.O.D. All authors have read and agreed to the published version of the manuscript.

**Funding:** This research was funded by the Society of Exploration Geophysicists (SEG) through the SEG/TGS field camp grant and the American Geophysical Union (AGU) Centennial grants. The APC was funded by the University of Toledo, Ohio, through research startup funds provided to Kennedy Doro.

**Data Availability Statement:** The data used to support the findings of this study are available from the corresponding author upon request.

**Acknowledgments:** The authors would like to thank students and participants in the 2019 SEG Field Camp and Field Workshop as apart of the Save-My-Water project, which were held at the research site. We thank Solomon Jekayinfa, Margaret Adeniran, Afolabi Adefehinti, and Ehinola for their support during the field exercises.

**Conflicts of Interest:** The authors declare no conflict of interest. The funders had no role in the design of the study; in the collection, analyses, or interpretation of data; in the writing of the manuscript; or in the decision to publish the results.

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
