# Peer review of "The Ibadan Hydrogeophysics Research Site (IHRS)—An Observatory for Studying Hydrological Heterogeneities in A Crystalline Basement Aquifer in Southwestern Nigeria"

_water, doi:10.3390/w15030433_

Round 1
Reviewer 1 Report
The objective of this paper is studying the hydrological heterogeneities in a crystalline basement aquifer in Southwestern Nigeria.
This is an interesting and well-structured paper, emphasizing research advances and knowledge gaps in crystalline basement aquifers in Nigeria, while all necessary sections (Introduction, The Ibadan Hydrogeophysics Research Site (IHRS), Materials and Methods, Discussion, Conclusions) are included. Moreover, almost all sections are divided into several sub-sections, providing detailed descriptions. Furthermore, all Figures, Diagrams and Tables are consistent with the analysis, described in the manuscript. However, some changes should be performed, which will result in the overall paper improvement. In particular:
Lines 11-30: The abstract is satisfactorily structured; however, many details have been included. The abstract should be brief and comprehensive. The additional details can be contained into the manuscript main body (not in the abstract). Please, modify.
Lines 87-96: This paragraph, presenting the paper objectives is vague; I suggest rephrasing it. Maybe numbering of the objectives can be performed. Please, apply.
Lines 99-100: The bibliographic references of the groundwater flow within crystalline rocks are poor and should be updated. Typical papers, in which significant information related to this subject, are contained in the following papers (they can optionally be cited): 1. Banks, D., Odling, N.E., Skarphagen, H., Rohr-Torp, E., 1996. Permeability and stress in crystalline rocks. Terra Nov. 8, 223–235. https://doi.org/10.1111/j.1365-3121.1996.tb00751.x, 2. Manoutsoglou, E., Lazos, I., Steiakakis, E., Vafeidis, A., 2022. The Geomorphological and Geological Structure of the Samaria Gorge, Crete, Greece—Geological Models Comprehensive Review and the Link with the Geomorphological Evolution. Appl. Sci. 12, 10670. https://doi.org/10.3390/app122010670, 3. Steiakakis, E., Monopolis, D., Vavadakis, D., Manutsoglu, E., 2011. Hydrogeological research in Trypali carbonate Unit (NW Crete), in: Advances in the Research of Aquatic Environment. Springer Berlin Heidelberg, Berlin, Heidelberg, pp. 561–567. https://doi.org/10.1007/978-3-642-19902-8_66, 4. Stober, Bucher, 1999. Origin of salinity of deep groundwater in crystalline rocks. Terra Nov. 11, 181–185. https://doi.org/10.1046/j.1365-3121.1999.00241.x. Please, apply.
Line 323: Please, provide Figure 2 in a higher resolution. It contains blur parts in the current form.
Line 519: The description of Figure 4 caption is poor. Please, provide a more detailed description.
Line 532: Similarly, the description of Figure 6 caption is poor. Please, provide a more detailed description.
Line 423: The “Conclusions” section should be modified. In the current form, it resembles an abstract rather than conclusions. This section should be comprehensive, while the major findings of the paper should be highlighted. Maybe, numbering of the conclusion remarks could be performed. Please, apply.
Author Response
Please see attached for response to the reviewer's comments.

Reviewer 2 Report
Line 178: replace “most of these research” with “most of these research efforts”
Lines 173-176: remove this section and highlight some research cases instead.
Line 186: “ID-ES” should be “1D-ES”
Figure 2: Legend for map A needs to be of a higher resolution.
Figure 3: Legend color needs to match colors on the map.
Line 342: Please include the meter model number and specs.
Line 366: Please include the model number and specs.
Line 376: what inversion code? Please elaborate.
Line 380: A, B, C and D on figure 3 are the same locations for both VES and test wells? Please explain.
Table 1. missing proper table caption.
Line 465: correct “ID” to “1D”
Figure 8: Fix the caption and remove “Figure 7:”
Figure 9 & 10: Change “Prole” to “Profile” on all transects.
Figure 9: Change “Prole” to “Profile” on all transects.
Line 511: please explain how you constructed the quasi 3D-resistivity model
Author Response
Please see attached for response to reviewer's comments.
